# Variation of Phenotypic Traits in Twelve Bambara Groundnut (*Vigna subterranea* (L.) Verdc.) Genotypes and Two F₂ Bi-Parental Segregating Populations

**Xiuqing Gao** [1], **Aliyu Siise Abdullah Bamba** [1,2,3], **Aloyce Callist Kundy** [1,4],
**Kumbirai Ivyne Mateva** [1,2], **Hui Hui Chai** [1,2], **Wai Kuan Ho** [1,2], **Mukhtar Musa** [1,2,5],
**Sean Mayes** [2,6,7] **and Festo Massawe** [1,2,*]

1 Future Food Beacon, School of Biosciences, University of Nottingham Malaysia, Jalan Broga,
 Semenyih 43500, Selangor Darul Ehsan, Malaysia; gaoxq0217@hotmail.com (X.G.);
 siisebamba@gmail.com (A.S.A.B.); ackundya@hotmail.com (A.C.K.); hbxkm1@nottingham.edu.my (K.I.M.);
 huihui.chai@nottingham.edu.my (H.H.C.); waikuan.ho@nottingham.edu.my (W.K.H.);
 mukhtar.musa@udusok.edu.ng (M.M.)
2 Crops for the Future, Jalan Broga, Semenyih 43500, Selangor Darul Ehsan, Malaysia;
 sean.mayes@nottingham.ac.uk
3 CSIR-Savannah Agriculture Research Institute, Off Tamale-Tolon Road, Nyankpala,
 Tolon District, Tamale P.O. Box TL 52, Ghana
4 Tanzania Agricultural Research Institute (TARI), Naliendele Centre, 10 Newala Road,
 Mtwara P.O. Box 509, Tanzania
5 Department of Crop Science, Usmanu Danfodiyo University, Sokoto P.M.B. 2346, Sokoto State, Nigeria
6 Plant and Crop Sciences, School of Biosciences, University of Nottingham, Sutton Bonington Campus,
 Leics, Loughborough LE12 5RD, UK
7 Crops for the Future (UK) CIC 76-80 Baddow Road, Chelmsford, Essex CM2 7PJ, UK
* Correspondence: festo.massawe@nottingham.edu.my

**Abstract:** Underutilised species such as bambara groundnut (*Vigna subterranea* (L.) Verdc.) have the potential to contribute significantly to meeting food and nutritional needs worldwide. We evaluated phenotypic traits in twelve bambara groundnut genotypes from East, West and Southern Africa and Southeast Asia and two F₂ bi-parental segregating populations derived from IITA-686 ×Tiga Nicuru and S19-3 ×DodR to determine phenotypic trait variation and their potential contribution to the development of improved crop varieties. All phenotypic traits in twelve genotypes were significantly influenced ($p < 0.01$) by genotypes. Principal component analysis (PCA) showed that PC1 accounted for 97.33% variation and was associated with four genotypes collected from East and Southern Africa. PC2 accounted for 2.48% of the variation and was associated with five genotypes collected from East, West and Southern Africa. Transgressive segregation for a number of traits was observed in the two F₂ bi-parental populations, as some individual lines in the segregating populations showed trait values greater or less than their parents. The variability between twelve genotypes and the two F₂ bi-parental segregating populations and the negative relationship between plant architectural traits and yield related traits provide resources for development of structured populations and breeding lines for bambara groundnut breeding programme.

**Keywords:** bambara groundnut; Bi-parental populations; plant architectural traits; yield related traits; plant breeding

## 1. Introduction

Bambara groundnut [*Vigna subterranea* (L.) Verdc.] is an underutilised legume crop, mainly grown by subsistence farmers in Africa [1–4]. Underutilised crops are still grown in their centres of origin or centres of diversity and are adapted to local conditions and marginal environments, while playing a significant role in food security, nutrition, income generation and cultural functions for people who grow them [2,5–9]. The potential of underutilised species to contribute to global food security and nutrition in the context of their significance for diversifying dietary and agriculture systems has been gaining prominence in recent times [2,6–8,10–12]. However, as the third most important food legume crop in semi-arid Africa after groundnut (*Arachis hypogaea* L.) and cowpea (*Vigna unguiculata* L. Walp) [13], bambara groundnut has often received limited support from government and international agencies [9,14,15].

The bambara groundnut plant life cycle is genotype dependent, ranging from 90 to 150 days and requiring 30 to 40 days to form pods after fertilisation, and generally reaches maturity most quickly under a photoperiod of 12 h [16,17]. Flowering in the bambara groundnut starts 30 to 45 days after planting and may continue until the end of life cycle, which is dependent on landraces and the environment [18]. The nodules formed on the roots fix atmospheric nitrogen, which is an important trait for soil fertility improvement and beneficial in crop rotation and intercropping [19]. The seed contains approximately 24% protein, 64% carbohydrates (53% starch, 10% dietary fibre), and 6% total fat, providing nutrition and a balanced diet for humans [20–22].

As with most of the underutilised and neglected crop species which lack established breeding programmes, landraces of bambara groundnut have remained the main source of planting materials used by farmers and the crop is still largely grown as landraces [14,23–26]. A major crop improvement programme is needed to enhance the genetic potential of bambara groundnut and to ensure sustainability and resilience, along with reasonable yield and quality. Development of improved varieties will bring into the market new materials and desirable traits such as early maturity, high yield and protein content, large pods and fast cooking to boost production and utilization of bambara groundnut [16,25]. Hybridization approaches for bambara groundnut have been reported and optimised [16,20,27]. The development of breeding resources that contribute towards variety development can also serve as material for genetic studies related to abiotic and biotic stress adaptive mechanisms.

In the present study, we characterised twelve genotypes from East, West and Southern Africa and Southeast Asia and two $F_2$ bi-parental segregating populations generated from four genotypes, IITA-686 ×Tiga Nicuru, S19-3 × DodR. This study aims to provide a better understanding of phenotypic trait variations among twelve genotypes collected from different geographical locations and the relationship between plant architectural traits and yield components, and to explore phenotypic trait variations in the segregating populations for potential contribution to crop variety development.

## 2. Materials and Methods

### 2.1. Plant Material and Growing Conditions

Twelve genotypes (developed by single plant descent (SPD) of bambara groundnut and two $F_2$ bi-parental segregating populations generated from four genotypes, IITA-686 × Tiga Nicuru (156 individual lines) and S19-3 × DodR (116 individual lines) were evaluated for phenotypic traits in a rainout shelter at the School of Biosciences, University of Nottingham Malaysia (2°56′46.74″ N; 101°52′24.35″ E) with mean 31 ± 4 °C/25 ± 1 °C day/night air temperature between February and June 2017. These twelve genotypes were collected from Africa and Southeast Asia, namely S19-3, Uniswa Red, DipC and AHM from Southern Africa, IITA-686, DodR and TAN385 from East Africa, LunT, Tiga Nicuru, Ankpa-4 and Getso from West Africa, Gresik from Southeast Asia (Table 1). Ten replicates of twelve genotypes were arranged in randomised complete block design (RCBD). Two $F_2$ bi-parental segregating populations, IITA-686 × Tiga Nicuru (156 individual lines) and S19-3 × DodR (116 individual lines) were planted in the separate plots with their parental lines.



The seeds of the twelve genotypes and the two segregating populations were soaked at room temperature (approximately 28 °C) for one day in the distilled water before sowing. A planting distance of 50 cm × 30 cm was established for the trials between the rows and between the plants. Fertiliser, including 1.86 kg nitrogen (N), phosphorus (P) and potassium (K) NPK (15:15:15) (133 kg/ha), 0.662 kg triple super phosphate (TSP) fertiliser (44 kg/ha) and 0.933 kg muriate of potash (MOP) (67 kg/ha) was applied two weeks after sowing. All the other agronomic procedures, such as watering, weeding and spraying of pesticides were carried out as and when necessary.

**Table 1.** Geographic origins and distinctive characteristics of twelve bambara groundnut genotypes.

| Geographical Origin | Landraces | Collected Country | Annual Rainfall (mm) | Distinctive Characteristics |
|---|---|---|---|---|
| East Africa | DodR | Tanzania | 1000 [28,29] | Quantitative for short days, high 100-seed weight and yield [27,30] |
| | IITA-686 | Tanzania | 1000 [28,29] | Quantitative for long days, shallow and highly branched root growth habit [27,31] |
| | TAN385 | Tanzania | 1000 [28,29] | - |
| West Africa | LunT | Sierra Leone | >2000 [31] | Quantitative for short days, shallow and highly branched root growth habit [27,31] |
| | Tiga Nicuru | Mali | 450 [28,32] | Quantitative for short days, bunchy growth habit, early maturity [27,33,34] |
| | Ankpa4 | Nigeria | >2000 [29,31] | Qualitative for short days [27] |
| | Getso | Nigeria | >2000 [29,31] | Quantitative for short days [27] |
| Southern Africa | S19-3 | Namibia | 365 [29,32] | Quantitative for long days, early maturity, drought tolerant long taproots and great root length distribution [27,31,32,34] |
| | Uniswa red | Kingdom of Eswatini | 1390 [32] | Quantitative for long days, long growth cycle [27,34] |
| | DipC | Botswana | 500 [31] | Quantitative for long days, long taproots and great root length distribution [27,31] |
| | AHM | Namibia | 365 [29,32] | - |
| Southeast Asia | Gresik | Indonesia | >2000 [31] | Quantitative for short days, shallow and highly branched root growth habit [27,31] |

## 2.2. Traits Recorded

Phenotypic traits, i.e., days to flowering, number of leaves per plant, petiole length, internode length, petiole internode ratio, plant height, 100-seed weight, harvest index and shelling percentage, were recorded based on the bambara groundnut descriptor list [35] with minor modification. Measurements included:

*Days to flowering, number of leaves per plant, petiole length, internode length* and *plant height. Petiole internode ratio* (P/I) was recorded based on the classification, Bunch type (P/I ≥ 9); Semi-bunch type (P/I = 7–9) and Spreading type (open) (P/I ≤ 7).

Yield data included *100-seed weight, harvest index* and *shelling percentage* recorded after pods were dried in a high-volume oven (Memmert, Germany) at 40 °C for 14 days.

## 2.3. Data Analysis

Analysis of variance (ANOVA), Tukey's multiple comparison test, W-test normality tests, principal components analysis (PCA) and Pearson's correlation coefficient tests were carried out for all phenotypic traits of twelve genotypes using 18th edition Genstat Statistical package (18th edition, VSN International, Hemel Hempstead, UK). Moreover, phenotypic traits of the two $F_2$ bi-parental segregating populations were subjected to Frequency distribution, Pearson's correlation coefficient tests and regression test using 18th edition Genstat Statistical package (18th edition, VSN International, UK).

## 3. Results

### 3.1. Phenotypic Trait Variation in Twelve Bambara Groundnut Genotypes

*Plant height, petiole length, internode length* and *100-seed weight* showed normal trait distribution in the twelve genotypes ($p > 0.05$). All phenotypic traits were significantly influenced ($p < 0.01$) by genotypes (Table 2). Comparing among twelve genotypes, Tiga Nicuru showed the earliest *days to flowering* the shortest *plant height, petiole length* and *internode length* while IITA-686 showed fewer *number of leaves per plant* but high *harvest index*. DodR was reported to have long *internode length*, high *number of leavers per plant, 100-seed weight*, while S19-3 had fewer *number of leaves per plant*, but high *harvest index* and *shelling percentage*. Growth habit ranged from bunch (LunT, Tiga Nicuru and Uniswa Red) to spreading types (DodR) and most genotypes classified as semi-bunch types (AHM, Ankpa4, Getso, Gresik, IITA-686, S19-3 and TAN385). The spreading growth habit type showed not only the longest *internode length* and but also the highest *plant height* in comparison to other growth habit types.

**Table 2.** Characterization of phenotypic traits in twelve genotypes of bambara groundnut.

| Traits | DTF | NL | PH | PL | IL | P/I | HI | 100SW | SP |
|--------|-----|-----|-----|-----|-----|-----|-----|-------|-----|
| | (days) | | (cm) | (cm) | (cm) | (ratio) | | (g) | (%) |
| AHM | 35.50 cd | 532.80 a | 35.20 a | 26.13 ab | 3.76 ab | 7.08 d | 0.09 cd | 21.45 efg | 77.70 a |
| Ankpa4 | 42.40 b | 198.20 b | 33.79 ab | 23.49 abc | 2.86 c | 8.23 cd | 0.04 d | 20.09 fg | 39.58 d |
| DipC | 38.57 bc | 250.40 b | 34.50 a | 27.04 a | 2.49 cde | 10.89 ab | 0.14 cd | 29.03 cdef | 75.10 a |
| DodR | 32.10 de | 266.50 b | 34.35 a | 21.60 c | 4.04 a | 5.55 e | 0.28 abc | 45.49 cdef | 78.01 a |
| Getso | 30.80 e | 82.10 c | 31.05 abc | 15.23 de | 1.83 ef | 8.49 cd | 0.16 bcd | 52.07 a | 62.74 bc |
| Gresik | 51.19 a | 159.60 b | 31.30 abc | 15.23 de | 2.11 de | 7.20 d | 0.26 abc | 33.32 cd | 74.54 a |
| IITA-686 | 32.50 de | 66.00 c | 28.15 bcd | 14.79 de | 2.09 de | 7.15 d | 0.38 ab | 24.68 def | 75.73 a |
| LunT | 33.00 de | 184.00 b | 34.44 a | 25.36 abc | 2.60 cd | 9.65 bc | 0.26 abc | 37.26 bc | 71.53 ab |
| S19-3 | 33.40 de | 84.80 c | 27.33 cd | 17.44 d | 2.46 cde | 7.30 d | 0.47 a | 27.47 cdef | 77.87 a |
| TAN385 | 33.50 de | 768.10 a | 34.50 a | 23.07 bc | 3.15 bc | 7.42 d | 0.03 d | 13.04 g | 61.74 c |
| Tiga Nicuru | 27.13 f | 81.10 c | 22.71 d | 12.52 e | 0.96 g | 13.36 a | 0.21 bcd | 24.58 def | 74.08 a |
| Uniswa Red | 33.44 de | 80.30 c | 28.33 bcd | 14.64 de | 1.13 fg | 13.24 a | 0.18 bcd | 31.05 cde | 73.21 a |
| Mean | 35.30 | 229.42 | 31.31 | 19.71 | 2.46 | 8.80 | 0.21 | 29.95 | 70.15 |
| F pr | <0.001 | <0.001 | <0.001 | <0.001 | <0.001 | <0.001 | <0.001 | <0.001 | <0.001 |

Note: DTF days to flowering, NL number of leaves per plant, PH plant height, PL petiole length, IL internode length, P/I petiole internode ratio, HI harvest index, 100SW 100-seed weight, SP shelling percentage. Different letters indicate significant difference at $p < 0.05$ level (Tukey's multiple comparison test), mean values ($n = 10$), F pr = F-probability ($p < 0.01$).

*Number of leaves per plant* ($r = 0.64$, $p < 0.05$), *plant height* ($r = 0.79$, $p < 0.01$) and *petiole length* ($r = 0.74$, $p < 0.01$) showed a strong linear relationship with *internode length* (Table 3). Both *number of leaves per plant* ($r = 0.61$, $p < 0.05$) and *plant height* ($r = 0.86$, $p < 0.01$) showed a strong linear relationship with petiole length. The negative correlation was observed between *harvest index* and *number of leaves per plant* ($r = -0.60$, $p < 0.05$). Yield related traits, namely *100-seed weight, harvest index, shelling percentage* showed a negative correlation with the key vegetative growth indices, *number of leaves per plant, plant height, petiole length, internode length* and *petiole internode ratio* ($r < 0.04$). A positive correlation between *harvest index* and *shelling percentage* ($r = 0.59$, $p < 0.05$) was observed.

**Table 3.** The correlation coefficient analysis of phenotypic traits in twelve genotypes of bambara groundnut.

| Traits | DTF | NL | PH | PL | IL | P/I | HI | 100SW | SP |
|--------|-----|-----|-----|-----|-----|-----|-----|-------|-----|
| DTF | - | - | - | - | - | - | - | - | - |
| NL | 0.05 | - | - | - | - | - | - | - | - |
| PH | 0.37 | 0.62 * | - | - | - | - | - | - | - |
| PL | 0.17 | 0.61 * | 0.86 ** | - | - | - | - | - | - |
| IL | 0.15 | 0.64 * | 0.79 ** | 0.74 ** | - | - | - | - | - |
| P/I | −0.28 | −0.35 | −0.50 | −0.27 | −0.77 ** | - | - | - | - |
| HI | −0.15 | −0.60 * | −0.49 | −0.45 | −0.18 | −0.18 | - | - | - |
| 100SW | −0.13 | −0.50 | 0.04 | −0.21 | −0.07 | −0.07 | 0.31 | - | - |
| SP | −0.23 | −0.12 | −0.25 | −0.18 | −0.04 | 0.03 | 0.59 * | 0.22 | - |

Note: DTF, days to flowering; NL, number of leaves per plant; PH, plant height; PL, petiole length; IL, internode length; P/I, petiole internode ratio; HI, harvest index; 100SW, 100-seed weight; SP, shelling percentage * = Significant at ($p = 0.05$), ** = Significant at ($p = 0.01$). Highlighted values represent highly significant correlation between phenotypic traits.

### 3.2. Principal Components Analysis for Twelve Genotypes Based on Phenotypic Traits

A principal component analysis (PCA) was carried out in order to investigate whether the trait variation observed among genotypes was influenced by the geographical locations where these genotypes were originally collected from. The first two principal components (PC), PC1 and PC2, accounted for 97.33% and 2.48% of the variation, respectively, with a cumulative variation of 99.81% (Figure 1). PC1 was associated with four genotypes collected from East and Southern Africa, i.e., TAN385 and DodR from East Africa and AHM and DipC from Southern Africa. The genotype TAN385 contributed 73% of the variation in the PC1. PC2 was associated with five genotypes from East, West and Southern Africa, i.e., IITA-686 from East Africa, Tiga Nicuru and Getso from West Africa, S19-3 and Uniswa Red from Southern Africa.

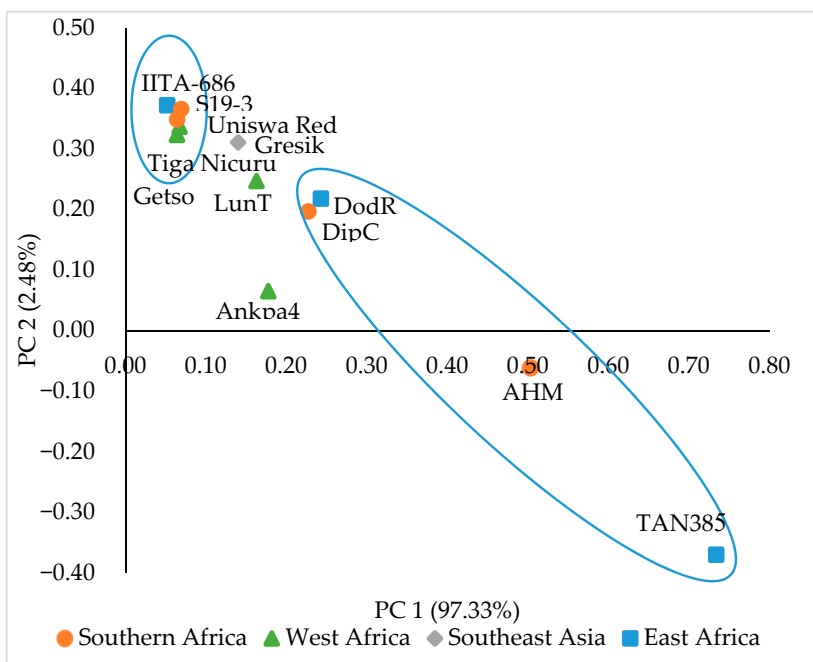

**Figure 1.** Principal component analysis (PCA) graph and loading scores for each component (PC1 and PC2) from latent vectors (loading), performed with Genstat Statistical package (18th edition, VSN International, Hemel Hempstead, UK) using phenotypic traits data of twelve genotypes. Data was coloured based on geographical collection origins.

### 3.3. Phenotypic Trait Variations in the F2 bi-Parental Segregating Populations

*Plant height, petiole length, 100-seed weight* and *harvest index* showed normal trait distribution in the $F_2$ bi-parental segregating population, IITA-686 × Tiga Nicuru and S19-3 × DodR ($p > 0.01$) (Tables 4 and 5). Transgressive segregation for traits was observed in the $F_2$ bi-parental segregating populations, IITA-686 × Tiga Nicuru and S19-3 × DodR (Figures 2 and 3). For example, the $F_2$ individual lines derived from IITA-686 × Tiga Nicuru had a *petiole length* that ranged from 11.83 cm (Line-66) to 30.67 cm (Line-125) whereas the *petiole length* in IITA-686 ranged from 10.10 cm to 19.75 cm (IITA-686 mean, 14.79 ± 0.87 cm; s.d. 2.73; $n = 10$) and Tiga Nicuru ranged from 10.60 cm to 15.57 cm (Tiga Nicuru mean, 12.52 ± 0.53 cm; s.d. 1.51, $n = 10$) (Table 4 and Figure 2). The $F_2$ individual lines derived from S19-3 × DodR had a *100-seed weight* that ranged from 10.79 g (Line-29) to 41.75 g (Line-96) whereas the *100-seed weight* in S19-3 ranged from 16.70 g to 39.58 g (S19-3 mean, 27.47 ± 2.95 g; s.d. 7.81; $n = 10$) and DodR ranged from 37.88 g to 52.70 g (DodR mean, 45.49 ± 1.57 g; s.d. 4.96, $n = 10$) (Table 5 and Figure 3).

**Table 4.** Summary phenotypic traits of the $F_2$ bi-parental segregating population derived from IITA-686 × Tiga Nicuru and their parental genotypes.

| | | | | | | | IITA-686 | | Tiga Nicuru | |
|---|---|---|---|---|---|---|---|---|---|---|
| Traits | Mean | Min | Max | SD | Variance | Normality | Min | Max | Min | Max |
| DTF | 34.98 | 31.00 | 61.00 | 3.54 | 12.51 | <0.001 | 29.00 | 34.00 | 27.00 | 28.00 |
| NL | 87.24 | 17.00 | 297.00 | 42.56 | 1811.00 | <0.001 | 28.00 | 114.00 | 43.00 | 116.00 |
| PH (cm) | 26.18 | 16.00 | 44.00 | 4.69 | 22.01 | 0.03 | 18.00 | 34.50 | 19.50 | 27.00 |
| PL (cm) | 19.72 | 11.83 | 30.67 | 3.50 | 12.23 | 0.13 | 10.10 | 19.75 | 10.60 | 15.57 |
| IL (cm) | 2.03 | 0.85 | 4.90 | 0.69 | 0.48 | <0.001 | 1.53 | 2.75 | 0.73 | 1.13 |
| P/I | 11.22 | 5.21 | 22.90 | 3.24 | 10.48 | 0.09 | 5.52 | 8.85 | 9.71 | 16.91 |
| HI | 0.24 | 0.01 | 0.46 | 0.11 | 0.01 | 0.45 | 0.06 | 0.76 | 0.06 | 0.36 |
| 100SW (g) | 21.85 | 4.69 | 45.56 | 7.70 | 59.31 | 0.57 | 15.67 | 29.50 | 19.90 | 35.21 |

Note: DTF, days to flowering; NL, number of leaves per plant; PH, plant height; PL, petiole length; IL, internode length; P/I, petiole internode ratio; HI, harvest index; 100SW, 100-seed weight. SD, standard deviation.

**Table 5.** Summary phenotypic traits of the $F_2$ bi-parental segregating population derived from S19-3 × DodR and their parental genotypes.

| | | | | | | | S19-3 | | DodR | |
|---|---|---|---|---|---|---|---|---|---|---|
| Traits | Mean | Min | Max | SD | Variance | Normality | Min | Max | Min | Max |
| DTF | 36.41 | 31.00 | 42.00 | 2.81 | 7.90 | <0.001 | 29.00 | 36.00 | 29.00 | 36.00 |
| NL | 77.67 | 15.00 | 325.00 | 50.61 | 2561.00 | <0.001 | 6.00 | 133.00 | 95.00 | 396.00 |
| PH (cm) | 28.34 | 13.30 | 46.50 | 5.66 | 32.02 | 0.02 | 18.30 | 33.00 | 27.00 | 44.00 |
| PL (cm) | 16.20 | 10.83 | 23.33 | 2.54 | 6.43 | 0.44 | 7.17 | 24.50 | 18.10 | 24.50 |
| IL (cm) | 2.48 | 0.73 | 4.50 | 0.77 | 0.59 | 0.55 | 1.17 | 3.55 | 3.00 | 5.00 |
| P/I | 7.09 | 3.89 | 16.36 | 2.15 | 4.64 | <0.001 | 5.63 | 9.37 | 3.81 | 7.50 |
| HI | 0.40 | 0.16 | 0.56 | 0.10 | 0.01 | 0.33 | 0.32 | 0.85 | 0.16 | 0.40 |
| 100SW (g) | 28.72 | 10.79 | 41.75 | 6.40 | 40.91 | 0.05 | 16.70 | 39.58 | 37.88 | 52.70 |

Note: DTF, days to flowering; NL, number of leaves per plant; PH, plant height; PL, petiole length; IL, internode length; P/I, petiole internode ratio; HI, harvest index; 100SW, 100-seed weight. SD, standard deviation.

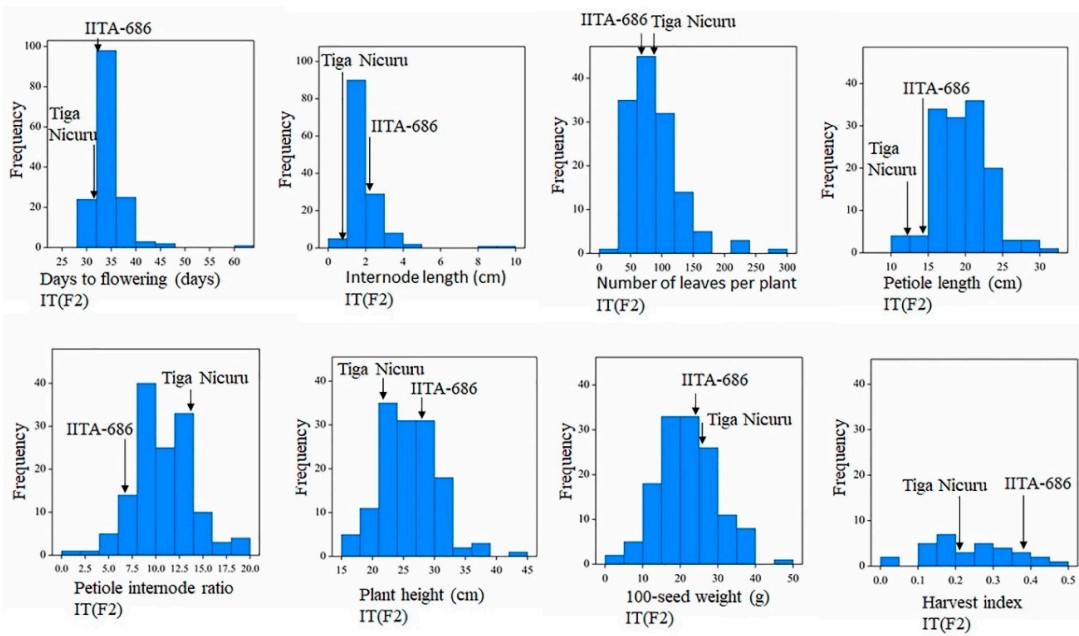

**Figure 2.** The frequency distribution of phenotypic traits in the $F_2$ bi-parental segregating population, IITA-686 × Tiga Nicuru and their parental lines. IT (F2), $F_2$ individual lines derived from IITA-686 × Tiga Nicuru.

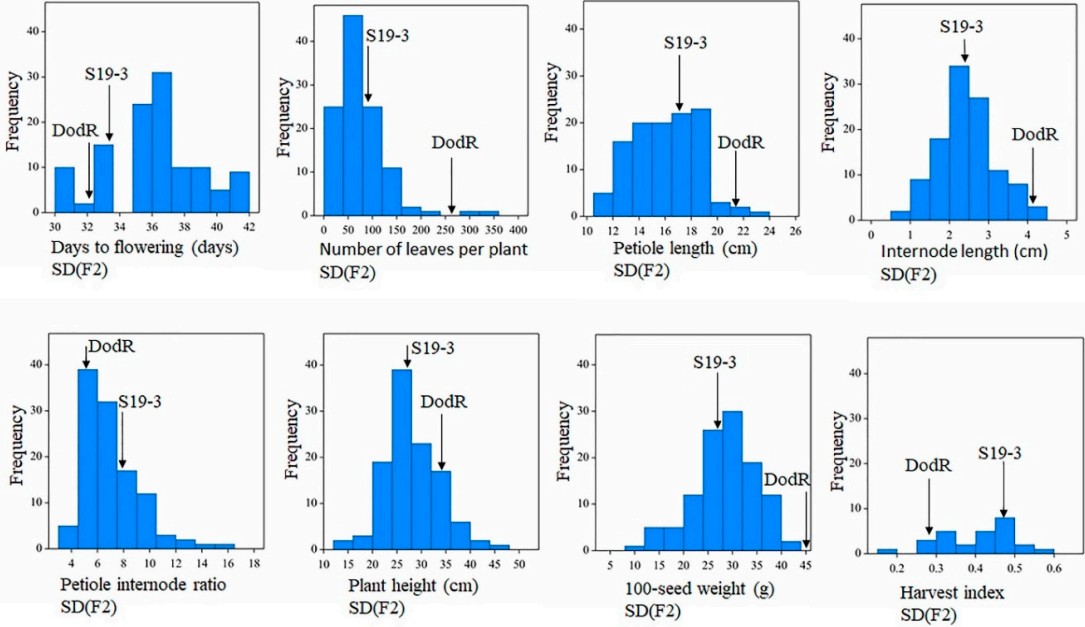

**Figure 3.** The frequency distribution of phenotypic traits in the $F_2$ bi-parental segregating population, S19-3 × DodR and their parental lines. SD (F2), $F_2$ individual lines derived from S19-3 × DodR.

*3.4. Correlation Coefficient Analysis of Phenotypic Traits in the $F_2$ bi-Parental Segregating Populations*

*Number of leaves per plant* showed positive correlation with *plant height* and *internode length* in the two $F_2$ bi-parental segregating populations, IITA-686 × Tiga Nicuru ($r = 0.54$, $p < 0.01$; $r = 0.54$, $p < 0.01$) and S19-3 × DodR ($r = 0.62$, $p < 0.01$; $r = 0.51$, $p < 0.01$) (Tables S1 and S2). *Harvest index* and *100-seed weight* showed strong positive linear relationship ($r = 0.81$, $p < 0.05$) in the $F_2$ segregating population derived from IITA-686 × Tiga Nicuru (Table S1, Figure 4A), and weak correlation ($r = 0.40$, $p < 0.05$) in the $F_2$ segregating population derived from S19-3 × DodR (Table S2).

*Harvest index* showed a negative correlation with the key vegetative growth indices, *number of leaves per plant* ($r = -0.50$, $p < 0.01$), *plant height* ($r = -0.70$, $p < 0.01$), *petiole length* ($r = -0.49$, $p < 0.01$), *internode length* ($r = -0.69$, $p < 0.01$) and positive correlation with *petiole internode ratio* ($r = 0.44$, $p < 0.05$) (Table S2). The negative linear relationships between *harvest index* and both *internode length* and *plant height* were observed in the $F_2$ segregating population derived from S19-3 × DodR (Figure 4B,C).

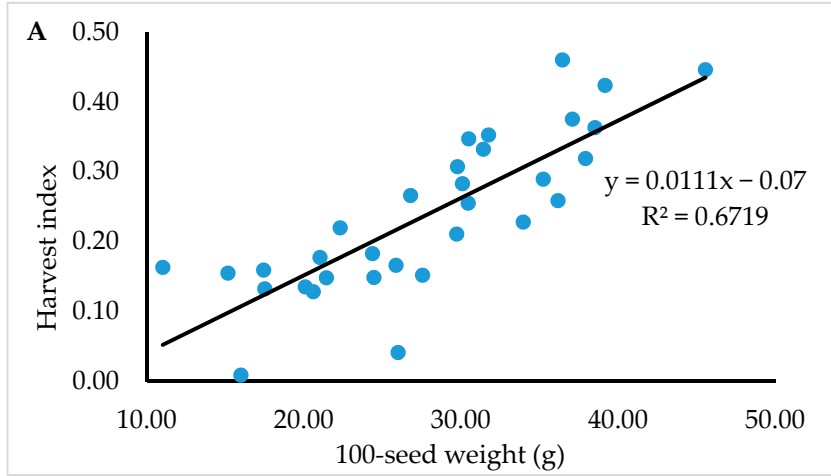

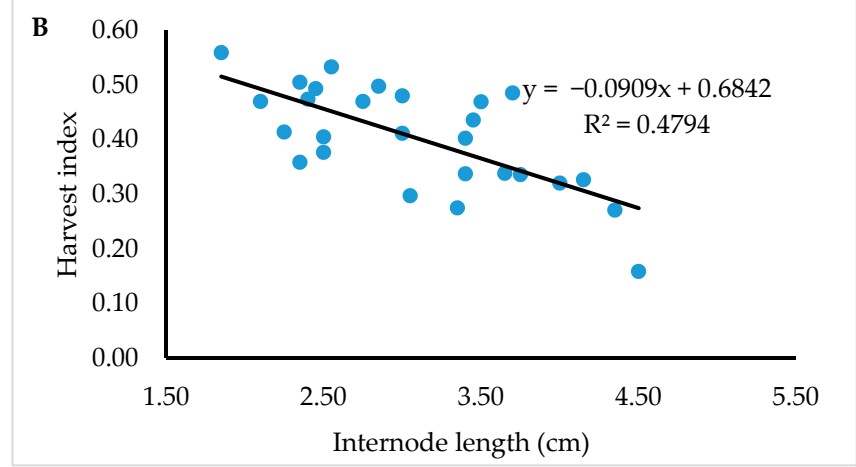

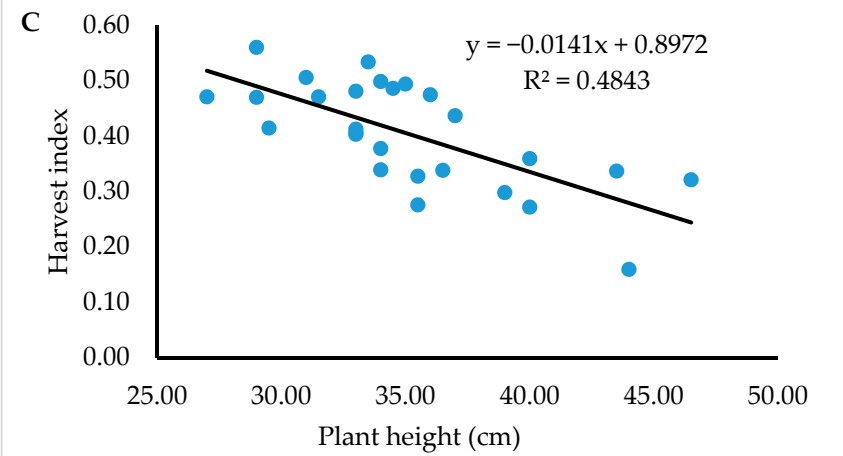

**Figure 4.** Regression for (**A**), harvest index and 100-seed weight (g) in the $F_2$ bi-parental segregating population derived from IITA-686 × Tiga Nirucru; (**B**), harvest index and internode length (cm) and (**C**), harvest index and plant height (cm) and in the $F_2$ bi-parental segregating population derived from S19-3 × DodR.

## 4. Discussion

The significance of single plant descent (SPD) in the context of deploying various short-to-medium term variety development strategies within bambara groundnut breeding programmes have been highlighted [16,36,37]. High inbreeding co-efficient and heterozygosity (Ho) below 5% observed in 119 landrace-derived genotypes (through SPD) of bambara groundnut studied by Molosiwa et al. [36] indicates that these cleistogamous landraces are likely to be composed of a series of inbred lines.

In the present study, PCA showed a total of 99.81% of the variation across the twelve genotypes based on days to flowering, number of leaves per plant, petiole length, internode length, petiole internode ratio, plant height, 100-seed weight, harvest index and shelling percentage, and the distribution was suggested to be related to geographic origins (Figure 1). PC1 was associated with high loadings in genotypes collected from moderate annual rainfall areas (600–1000 mm annual rainfall) and semi-arid areas (200–600 mm annual rainfall) in Africa, including TAN385 and DodR from Tanzania, East Africa with 1000 mm mean annual rainfall, and AHM from Namibia and DipC from Botswana, Southern Africa with less than 600 mm mean annual rainfall [28,29]. PC2 was associated with high loadings in genotypes from different origins with semi-arid areas, moderate and high (above 1000 mm annual rainfall) annual rainfall in East, West and Southern Africa. Tiga Nicuru was collected from Mali, West Africa and S19-3 was collected from Namibia, Southern Africa with less than 450 mm mean rainfall per year [29,32]. IITA-686 was collected from Tanzania, East Africa with 1000 mm mean annual rainfall [28,29]. Getso collected from northern Nigeria, West Africa with more than 2000 mm mean annual rainfall [29,31]. Uniswa Red was collected from the Kingdom of Eswatini, Southern Africa with 1390 mm mean annual rainfall [32]. Furthermore, the high genotypic variability between landraces of bambara groundnut allows breeders to select parents for controlled crossing to develop and release new improved varieties with desirable traits. S19-3, TAN385, ZAM696, AHM753 and BOTS1 were recommended as the best performing genotypes with good yield component traits in Botswana [36,38]. Tiga Nicuru and S19-3 are likely to avoid terminal drought stress by early maturity or reduced respiration and stomata closure at a comparatively lower water threshold coupled with fast phenological development [14,32–34,39,40] and longer tap roots, as well as greater root length distribution in deeper (60–90cm) soil depths [31]. The variation of genotypes provides opportunities to develop ideotypes with drought tolerant, high yield, short life cycle or other favourable traits in breeding programmes of bambara groundnut.

In the present study, the negative correlation between the plant architectural traits, i.e., *number of leaves per plant* (NL), *petiole length* (PL), *internode length* (IL), *petiole internode ratio* (P/I), *plant height* (PH) and yield-related traits, i.e., *100-seed weight* (100SW), *harvest index* (HI) and *shelling percentage* (SP), would suggest that fewer leaves, reduced PL and IL and shorter plants could lead to high 100SW and HI. Similar findings have been reported in pea (*Pisum sativum* L.) that the increased generative shoots and fruiting nodes had a negative impact on the harvest index [41]. Furthermore, plant height was negatively correlated with a key yield component, number of tillers in a bushy rice mutant [42]. Flowering time, typically after the vegetative stage, is a decisive trait for yield improvement in crop plants under different environmental conditions [43]. Mabhaudhi et al. [44] reported that early flowering to escape drought stress can lead to early maturity but has a yield penalty (reduced seed yield) in bambara groundnut. However, no significant correlation between flowering time and other phenotypic traits was observed in the twelve genotypes and two $F_2$ segregating populations in the present study. From the present study and previous report [45], the semi-bunch growth type was the most common growth habit type (58.3%), followed by bunch (33.3%) and spreading (8%) (Table 2). The significantly negative correlation between *internode length* and *petiole internode ratio* in the present study, suggests internode length is the most critical trait to determine the plant growth habit type in bambara groundnut, similar to Basu et al. [23].

The two sets of parental genotypes in the present study had contrasting traits and were selected for controlled crossing to develop two segregating populations for the selection of breeding lines for variety development and to act as mapping populations for genetic studies. Results showed high

variability and transgressive segregation in the $F_2$ bi-parental segregating lines [40]. Transgressive segregation identified for trait values including *days to flowering, number of leaves per plant, petiole length, internode length, plant height, petiole internode ratio, 100-seed weight, harvest index*, and *shelling percentage* in the two $F_2$ bi-parental segregating populations, provides an opportunity for selection of superior individuals for breeding purposes. The significant and negative correlation between *harvest index, plant height* and *internode length* in the $F_2$ segregating population, S19-3 × DodR (Table S2), which was also confirmed by regression analysis (Figure 4B,C), suggested the possibility of developing individual lines with high yield through selection of target genotypes with short internode or height for breeding improvement. For example, Some individual lines, i.e., Line-6, 38, 44, 48, 50 in the $F_2$ bi-parental segregating population, IITA-686 × Tiga Nicuru and Line-51, 73, 86, 108, 111 in the $F_2$ bi-parental segregating population, S19-3 × DodR, combined with desirable traits, such as earlier flowering, higher *harvest index, 100-seed weight*, shorter *plant height* and *internode length* than average of the population are recommended for further field investigation to develop improved varieties (Tables S3 and S4). However, it is worth taking note that final yield is rather complex due to the possible interaction between genetic and environmental factors, which could contribute to the high variability observed between genotypes and within segregating lines [38,40,46]. As offspring segregate for agronomically important traits, breeders can select target lines that are adapted to the target environments based on their breeding and selection plan. In addition to selection of lines, the development of structured populations and breeding lines provides resources for genetic analysis and trait dissection, i.e., genetic mapping and identification of regions of the genome correlated with phenotypic traits.

## 5. Conclusions

The present study provides initial results from the two $F_2$ structured populations and clears a path to develop the first ever advanced structured populations and improved varieties of bambara groundnut. The variation within twelve genotypes of bambara groundnut provides a breeding resource pool for use in controlled crossing to develop ideotype varieties with desirable phenotypic traits, i.e., high *harvest index, 100-seed weight,* early *days to flowering* or short life cycle. Two $F_2$ bi-parental segregating populations of bambara groundnut derived from different geographical origins, IITA-686 (high *harvest index*, collected from a moderate annual rainfall area, Tanzania, East Africa) × Tiga Nicuru (early *days to flowering*, collected from semi-arid area, Mali, West Africa) and S19-3 (high *harvest index*, collected from semi-arid area, Namibia, Southern Africa,) × DodR (high *100-seed weight*, collected from moderate annual rainfall area, Tanzania, East Africa), were developed to obtain structured populations and breeding lines for genetic analysis and trait dissection. The negative correlation between the plant architectural traits, i.e., *number of leaves per plant, petiole length, internode length, petiole internode ratio* and *plant height* and yield-related traits, i.e., *100-seed weight, harvest index* and *shelling percentage*, would suggest a competition for assimilates between vegetative development and yield accumulation. Therefore, a balanced development of vegetative growth and yield accumulation is a critical strategy to obtain improved varieties for breeding programme. Individual lines in the segregating populations with higher *harvest index* or *100-seed weight*, earlier flowering and shorter *plant height* and *internode length* could be selected as potential high yield genotypes for improved variety development. Further studies would focus on advanced generations to investigate the correlation between plant architectural traits, yield-related traits and final yield and to identify potential genomic regions involved in the regulation of key agronomic traits in bambara groundnut.

**Supplementary Materials:** The following are available online at http://www.mdpi.com/2073-4395/10/10/1451/s1, Table S1: Correlation coefficient analysis of phenotypic traits in the $F_2$ bi-parental segregating population derived from IITA-686 × Tiga Nicuru. Table S2: Correlation coefficient analysis of phenotypic traits in the $F_2$ bi-parental segregating population derived from S19-3 × DodR. Table S3: Potential lines (cross IITA-686 × Tiga Nicuru) with superior performance than population mean for advancement based on days to flowering, harvest index and 100-seed weight, plant height and internode length. Table S4: Potential lines (cross S19-3 × DodR) with superior performance than population mean for advancement based on days to flowering, harvest index and 100-seed weight, plant height and internode length.

**Author Contributions:** Conceptualization, X.G. and F.M.; methodology, X.G., H.H.C. and F.M.; formal analysis, X.G.; investigation, X.G., A.S.A.B., A.C.K., and M.M.; writing—original draft preparation, X.G.; writing—review and editing, X.G., A.S.A.B., K.I.M., H.H.C., W.K.H., S.M. and F.M.; supervision, F.M. and H.H.C.; funding acquisition, F.M. and S.M. All authors have read and agreed to the published version of the manuscript.

**Funding:** This work has been supported and funded by The University of Nottingham Malaysia and the Crops for the Future—Nottingham Malaysia Campus Doctoral Training Partnership (CFF-UNMC DTP) programme.

**Conflicts of Interest:** The authors declare no conflict of interest.

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
