# Peer review of "Variation of Phenotypic Traits in Twelve Bambara Groundnut (Vigna subterranea (L.) Verdc.) Genotypes and Two F2 Bi-Parental Segregating Populations"

_agronomy, doi:10.3390/agronomy10101451_

Round 1
Reviewer 1 Report
The paper entitled “Variation of phenotypic traits in twelve bambara groundnut (Vigna subterranea (L.) Verdc.) genotypes and two F2 bi-parental segregating populations” by Xiuqing Gao, Aliyu Siise Abdullah Bamba, Aloyce Callist Kundy, Kumbirai Ivyne Mateva, Hui Hui Chai, Wai Kuan Ho, Sean Mayes, and Festo Massawe presents agronomy traits which might be relevant for the selection and breeding of Bambara groundnut varieties to improve food security in the areas it is grown. The experiment was well designed but unfortunately it was performed only once and should be repeated at least a second year to ensure the significance of the results. The English in the text should be improved and Tukey’s test was not mentioned in the Material and Methods part. The paper should be rewritten by specifying that it is only preliminary results that need more data to be validated/significant because as the authors mentioned, this type of underutilised crops deserves more research credits.
Reviewer 2 Report
The study entitled "Variation of phenotypic traits in twelve bambara groundnut (Vigna subterranea (L.) Verdc.) genotypes and two F2 bi-parental segregating populations" aims to provide a better understanding of phenotypic trait variations among twelve genotypes collected from different geographical locations, the relationship between plant architectural traits and yield components, and to explore phenotypic trait variations in the segregating populations for potential contribution to crop variety improvement.
The trial is carried out for one year.
The introduction provides a good background for the topic, but the objectives of breeding programs should be better explained. What new desirable traits do consumers and farmers look for in the new Bambara peanut varieties?
Material and methods must be well explained by adding also a table with the origin of the twelve genotypes and their distinctive characteristics.
Finally in discussion the most interesting lines for the two crosses should be described in a table.
Round 2
Reviewer 1 Report
The revision improves the quality of the manuscript, with the new Table 1 a good addition to it.
The authors took into account my suggestions, even though it might have been better to add more than just one line at the beginning of the conclusion of the limitations of a one-year study. But as I said before, this paper deserves to be published as it will be a good addition to the literature about this crop.
One minor suggestion (maybe for another paper) is to use heat maps to present correlation tables (see example attached), and maybe bold number for significant correlations.
